# A mutation-level covariate model for mutational signatures

**Itay Kahane** [1], **Mark D. M. Leiserson**[2], **Roded Sharan**[1]*

**1** School of Computer Science, Tel Aviv University, Tel Aviv, Israel, **2** Department of Computer Science and Center for Bioinformatics and Computational Biology, University of Maryland, College Park, Maryland, United States of America

* roded@tauex.tau.ac.il

## Abstract

Mutational processes and their exposures in particular genomes are key to our understanding of how these genomes are shaped. However, current analyses assume that these processes are uniformly active across the genome without accounting for potential covariates such as strand or genomic region that could impact such activities. Here we suggest the first mutation-covariate models that explicitly model the effect of different covariates on the exposures of mutational processes. We apply these models to test the impact of replication strand on these processes and compare them to strand-oblivious models across a range of data sets. Our models capture replication strand specificity, point to signatures affected by it, and score better on held-out data compared to standard models that do not account for mutation-level covariate information.

## Author summary

Somatic mutations, caused by processes such as DNA damage and faulty DNA repair, may lead to cancer. Studying the mutational signatures those processes leave behind, provides insights on their activities and can be utilized for personalized therapy. Previous methods for analyzing mutational signatures did not account for the fact that some signatures tend to occur in varying frequencies along the genome, depending on positional factors such as strand identity or genomic region. In this work, we develop new models that account for these factors, and show that exploiting such information improves the inference of mutational signatures and their activities with applications to both basic science and personalized medicine.

## 1 Introduction

Cancers are caused by somatic mutations accumulated during the organism's life [1, 2]. Those mutations, are the result of mutational processes varying from exogenous and endogenous DNA damage to faulty DNA repair and replication [3, 4], and leaving unique mutational signatures [1, 2, 5]. Deciphering these signatures and the genome's exposure to them are key to understanding how it is shaped by the disease. Such mapping was initially done by non-

**Data Availability Statement:** The data and code are available by a link given in the manuscript. It can also be found on the following link: https://github.com/Kitay10/Mutation-level-covariate-models.

**Funding:** This research was supported by a grant from the United States - Israel Binational Science Foundation (BSF), Jerusalem, Israel (to RS and MDML). RS was further supported by a grant on collaborative clinical bioinformatics research of the Edmond J. Safra Center for Bioinformatics at Tel Aviv University and Sheba Cancer Center at Sheba Medical Center. The funders had no role in study design, data collection and analysis, decision to publish, or preparation of the manuscript.

negative matrix factorization (NMF) and its generalizations [1, 6–9], or refitting methods that infer the exposures given the signatures [10–14]. More recent work built on topic models that allow to rigorously attribute likelihood to the data and solve the models' parameters by maximizing it [15–19]. One of the advantages of the topic model framework, is that it allows to exploit additional information on the data for improved predictions. For instance, [17] used a generalization of Dirichlet multinomial regression [20] to introduce tumor-level covariates in the context of mutational signature modeling.

The aforementioned methods assume that mutational processes work uniformly across the genome. However, it was previously reported that some mutational processes have strand [21–23] and region [24] biases. In particular, signatures 2, 13 and 26 (following COSMIC v2 catalogue [25]) were found to have strong replication strand biases [22].

Here we suggest the first mutation-covariate models that explicitly model the effect of different covariates on the exposures of mutational processes. We apply these models to test the impact of genomic and replication strands on these processes and compare them to strand-oblivious models across a range of data sets.

## 2 Results

We designed novel models for mutational signature analysis that account for mutation-level features. The basic model, MCSM, is a generalization of the standard LDA that learns two different Dirichlet priors for the case of a binary feature, each corresponding to a different value of the feature. A refined model, JMCSM, accounts also for tumor-level covariates in the form of a vector exposures that ties together the two Dirichlet priors (with an LDA-like benchmark termed gLDA). To test our models we focused on the richest features that we could extract, namely the genomic and replication strand information across different data sets. We applied the models in a refitting setting where signatures are given as input and their exposures have to be inferred, and compared each feature-sensitive model with its feature-oblivious analogue, i.e MCSM with LDA and JMCSM with gLDA. While Watson/Crick data did not seem to improve learning (see Conclusions and Table A In S1 Text), notable differences were observed with respect to the lagging/leading strand data (Table 1).

As evident from the table, the refined JMCSM and gLDA models yield better held-out log-likelihood than their basic versions thanks to their usage of covariate information of *inherent exposures*. While MCSM does not consistently improve upon LDA, JMCSM dominates the other models when tested on the larger datasets, BRCA and MALY.

Next, focusing on JMCSM, we wished to pinpoint the signatures with replication strand bias. To this end, we calculated for each signature the log-ratio magnitude of its normalized modification parameters $a_k/\sum_i a_i$ and $b_k/\sum_i b_i$. As $a$ and $b$ indicate the relative bias of a signature given a feature, the normalized ratio indicates its intensity. The results are given in Fig 1 for the two larger datasets.

It can be seen that although Signatures 2, 13, 26 are known to have a strong replication strand bias [22], some other signatures seem to have stronger bias in our framework, most notably Signature 6 which has the highest log-ratio among all tested signatures.

**Table 1. A comparison between MCSM and LDA, and between JMCSM and gLDA using replication strand as a mutation-level feature.** The percentage difference sign indicates whether the strand sensitive model has a better likelihood (negative), and vice versa.

| dataset | MCSM | LDA | diff % | JMCSM | gLDA | diff % |
|---------|------|-----|--------|-------|------|--------|
| BRCA | -2,216,362 | -2,215,837 | 0.024 | -1,995,494 | -1,996,589 | -0.055 |
| MALY | -717,961 | -718,491 | -0.074 | -694,699 | -694,937 | -0.034 |
| CLLE | -112,664 | -112,703 | -0.035 | -111,869 | -111,866 | 0.002 |

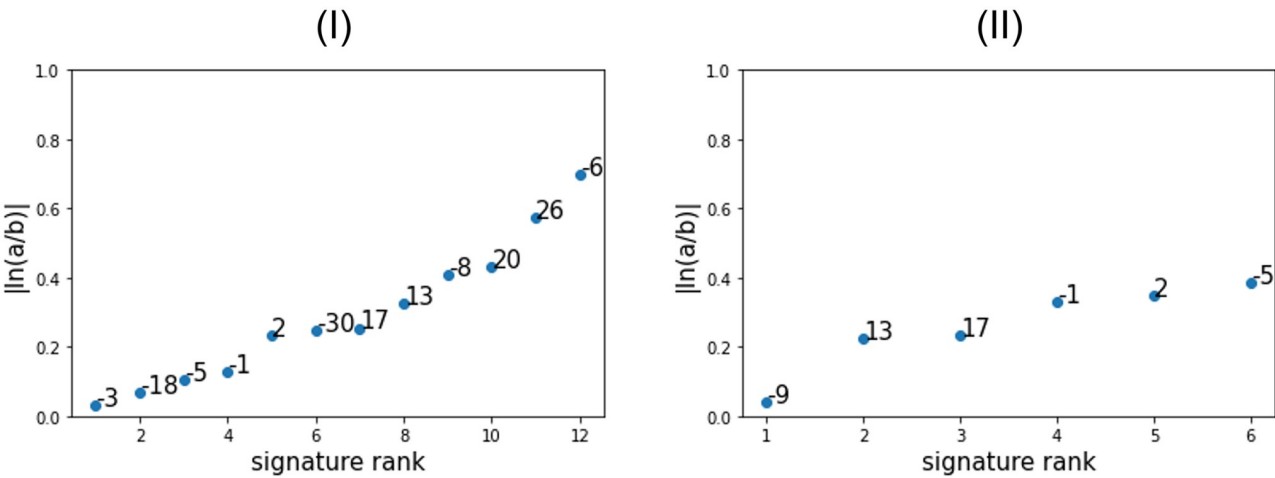

**Fig 1. Log-ratio magnitude of *a* and *b* for each signatures in BRCA (I) and MALY (II).**

A major disadvantage of this method is that it does not take into account the signature frequency in the dataset. A rare signature would have a relatively small effect on the held-out likelihood, and its respective *a* and *b* bias parameters are more prone to overfitting. Also, since the normalized parameter vector represents compositional data, signature-specific results are hard to interpret. To tackle those problems, we used a second evaluation of strand bias by calculating the contribution of each signature seperately to the held-out log likelihood. To do so, we compared the likelihood obtained when imposing no bias on the signatures to the likelihood obtained when allowing only a single signature to be biased. Once the modification parameters *a* and *b* are estimated, we do it by flattening the parameters respective to the unbiased signatures to their average such that $a_k \rightarrow (\sum_i a_i - a_q)/(K - 1)$ and $b_k \rightarrow (\sum_i b_i - b_q)/(K - 1)$, where $q$ is the biased signature. More intuitively, since the modification parameters are very high ($\sim 1,000 - 10,000$), the probability vector drawn from the Dirichlet distribution is very close to the distribution parameters, resulting in an exposure ratio determined only by the *inherent exposures*. The results are given for BRCA and MALY in Fig 2.

The results indicate that for BRCA the main sources for the improvement in likelihood are signatures 6, 8, 13 and 26 (the last two are known to have a strong replication strand bias, as stated earlier). Signature 20 (which is also the rarest in BRCA, in terms of exposure averaged over all samples), however, does not seem to display a positive effect on the likelihood contrary to its high bias using the first evaluation. Considering MALY, signatures 2, 5 and 17 are the major likelihood contributors. Signature 13, despite its strong replication strand bias and its impressive contribution in the BRCA case, yields smaller contribution due to the fact that most of the mutations in the database are assigned more likely to other signatures.

## 3 Discussion

We have developed new topic models to account for mutation-level covariates, focusing on a refitting scenario where signatures are known and their exposures are to be learned. The models allowed us to pinpoint signatures that display replication strand bias. Notably, we also applied our models with the Watson/Crick strand of the pyrimidine base in the reference sequence as a feature but could not detect any advantage of the strand-sensitive models over their strand-oblivious counterparts, and indeed such biases are not reported in the literature. While we applied our framework in the context of strand information, the models described

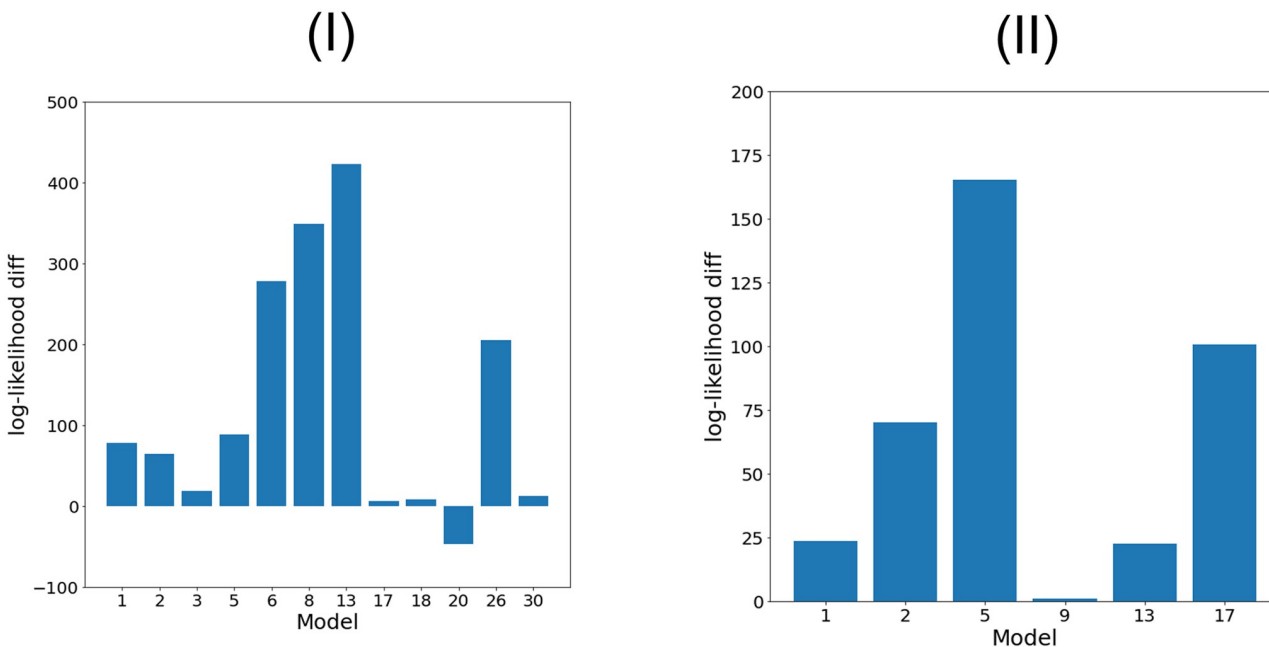

**Fig 2. Held-out log-likelihood contribution of each signature for BRCA (I) and MALY (II).** The y-axis shows the difference between the log-likelihood of an unbiased model and a single signature bias model. The number under the bars indicates the signature whose bias is maintained.

here can be easily adapted to other binary or categorial features, and could perhaps be used to reveal other biological differences between mutations, for example when considering coding vs. non-coding regions.

## 4 Methods

### 4.1 Preliminaries

We follow the common convention and assume that there are $M = 96$ mutation categories (denoting a base substitution and its flanking bases), drawn from $K$ known signatures. Our data consists of $T$ samples, each of which is a set of somatic mutations along with category and feature information per mutation. We assume that the mutational signatures that are active in the given dataset are known. For simplicity, we focus on a single binary feature or covariate per mutation such as the genomic strand on which it occurred. The mutation category count of the two possible feature values for each sample $t$ are denoted by $I_{t,m}$ and $J_{t,m}$, respectively.

The basic model we consider, which can be thought of as the probabilistic analog of NMF, is the multinomial mixture model (MMM) [18]. In MMM, the signatures are modeled as multinomial distributions, such that the probability to draw a mutation $m$ from signature $k$ is notated by $\gamma_{k,m}$. Further, each signature has sample-specific probabilities, aka *exposures*. The model specifies a generative process for mutations, where at first a signature is drawn from an exposure vector. Then, a mutation category is drawn from the signature vector (see Fig 3 for a plate notation). A shortcoming of this model is the fact that exposure vectors of different samples are assumed to be independent.

To mitigate the latter drawback and generalize to unseen samples, Latent Dirichlet allocation (LDA) assumes that the samples share an exposure Dirichlet prior, rather than the exposure vector itself [26]. First, an exposure vector $\theta$ is drawn per-sample from a Dirichlet prior. Then, as in MMM, a signature $z$ is drawn from the multinomial distribution

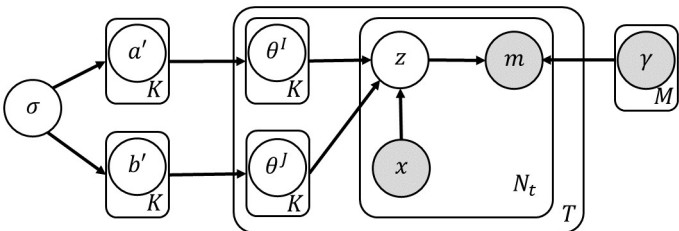

**Fig 3. A plate notation of LDA.** The conditional distributions of each node are given by: $a'_k \sim Norm(0, \sigma^2)$; $\theta \sim Dir(a)$; $z \sim Multi(\theta)$; and $m \sim Multi(\gamma_z)$.

specified by the exposure vector and a mutation category $m$ is drawn from the multinomial distribution specified by $z$. Let $a'_k \sim Norm(0, \sigma^2)$ and denote $a_k = \exp(a'_k)$, $A = \sum_{k=1}^{K} a_k$. Further denote by $N_{t,k}$ the number of times signature $k$ occur in sample $t$ and $N_t = \sum_{k=1}^{K} N_{t,k}$. Then the likelihood of the model when marginalizing over all exposure vectors and omitting the mutation category emissions is given by:

$$\mathcal{L}_{LDA}(\underline{a}) = \left[ \prod_{t=1}^{T} \frac{\Gamma(A)}{\Gamma(A + N_t)} \left( \prod_{k=1}^{K} \frac{\Gamma(a_k + N_{t,k})}{\Gamma(a_k)} \right) \right] \left( \prod_{k=1}^{K} \frac{1}{\sigma\sqrt{2\pi}} exp\left(-\frac{a'^2_k}{2\sigma^2}\right) \right) \tag{1}$$

## 4.2 Mutation-level covariate signature model (MCSM)

Similar to Dirichlet multinomial regression, we can generalize the above scheme to model mutation-level features such as strand identity assuming each sample has two exposure vectors (one for each strand), and that each mutation's signature assignment is drawn from the respective exposure vector, given its strand feature (see Fig 4 for a plate notation). In this model, the binary feature denoted by $x$ determines the parameters of the Dirichlet distribution from which the exposures are drawn. Using similar notation as before, with $b$ in addition to $a$ for the second feature and $I$ and $J$ instead of $N$ for the feature-divided data, the likelihood of the signature part of the model is given by:

$$\mathcal{L}_{MCSM}(\underline{a}, \underline{b}) = \left[ \prod_{t=1}^{T} \frac{\Gamma(A)}{\Gamma(A + I_t)} \left( \prod_{k=1}^{K} \frac{\Gamma(a_k + I_{t,k})}{\Gamma(a_k)} \right) \right] \left( \prod_{k=1}^{K} \frac{1}{\sigma\sqrt{2\pi}} exp\left(-\frac{a'^2_k}{2\sigma^2}\right) \right)$$
$$\left[ \prod_{t=1}^{T} \frac{\Gamma(B)}{\Gamma(B + J_t)} \left( \prod_{k=1}^{K} \frac{\Gamma(b_k + J_{t,k})}{\Gamma(b_k)} \right) \right] \left( \prod_{k=1}^{K} \frac{1}{\sigma\sqrt{2\pi}} exp\left(-\frac{b'^2_k}{2\sigma^2}\right) \right) \tag{2}$$

**Fig 4. A plate notation of MCSM.** The conditional distributions of each node are given by: $a'_k, b'_k \sim Norm(0, \sigma^2)$; $\theta^I \sim Dir(a)$; $\theta^J \sim Dir(b)$; $z \sim Multi(\theta^I)|x = 0$, $z \sim Multi(\theta^J)|x = 1$; and $m \sim Multi(\gamma_z)$.

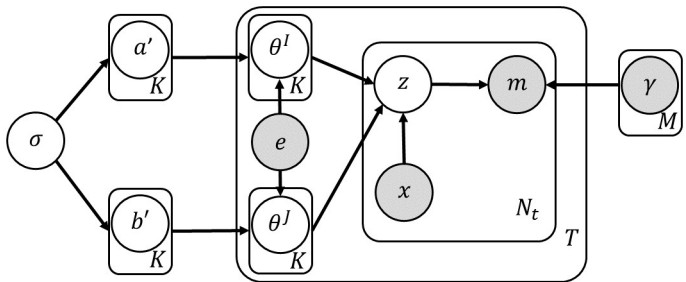

**Fig 5. A plate notation of JMCSM.** The conditional distributions of each node are given by: $a'_k, b'_k \sim Norm(0, \sigma^2)$; $\theta^I \sim Dir(e \odot a)$; $\theta^J \sim Dir(e \odot b)$; $z \sim Multi(\theta^I)|x = 0$, $z \sim Multi(\theta^J)|x = 1$; and $m \sim Multi(\gamma_z)$.

### 4.3 Joint MCSM (JMCSM)

Although the previous MCSM model allows us to integrate mutation level features, it assumes that every tumor has two independent exposure vectors. In reality, since signature exposures are the consequences of genetics and lifestyle, it is reasonable to assume that the two exposures vectors are related rather than independent. To capture this dependency, we assume that a tumor $t$ has an *inherent exposure vector* denoted by $e_t = (e_{t,1}, \ldots, e_{t,K})$ which can be thought of as a tumor level covariate [17, 20] (see Fig 5 for a plate notation). In order to impose that the strand-specific exposure vector $\theta^I$ or $\theta^J$ is drawn in the proximity of the inherent exposures, we modify the Dirichlet parameters and define them as $(e_{t,1}a_1, \ldots, e_{t,K}a_K)$ and $(e_{t,1}b_1, \ldots, e_{t,K}b_K)$. The likelihood of the signature part of the model is now given by:

$$
\mathcal{L}_{JMCSM}(\underline{a}, \underline{b}) =
$$
$$
\left[ \prod_{t=1}^{T} \frac{\Gamma(\sum_k e_{kt} a_k)}{\Gamma(\sum_k e_{kt} a_k + I_t)} \left( \prod_{k=1}^{K} \frac{\Gamma(e_{kt} a_k + I_{t,k})}{\Gamma(e_{kt} a_k)} \right) \right] \left( \prod_{k=1}^{K} \frac{1}{\sigma\sqrt{2\pi}} exp\left(-\frac{a_k'^2}{2\sigma^2}\right) \right)
$$
$$
\left[ \prod_{t=1}^{T} \frac{\Gamma(\sum_k e_{kt} b_k)}{\Gamma(\sum_k e_{kt} b_k + J_t)} \left( \prod_{k=1}^{K} \frac{\Gamma(e_{kt} b_k + J_{t,k})}{\Gamma(e_{kt} b_k)} \right) \right] \left( \prod_{k=1}^{K} \frac{1}{\sigma\sqrt{2\pi}} exp\left(-\frac{b_k'^2}{2\sigma^2}\right) \right)
\tag{3}
$$

To tease apart the contribution of the joint modeling of strands on model performance, we also define a guided LDA (gLDA) model variant which is fed with external information on the exposures as in JMCSM (see Fig 6 for a plate notation). Its likelihood is given by:

$$
\mathcal{L}_{gLDA}(\underline{a}) =
$$
$$
\left[ \prod_{t=1}^{T} \frac{\Gamma(\sum_k e_{kt} a_k)}{\Gamma(\sum_k e_{kt} a_k + I_t)} \left( \prod_{k=1}^{K} \frac{\Gamma(e_{kt} a_k + I_{t,k})}{\Gamma(e_{kt} a_k)} \right) \right] \left( \prod_{k=1}^{K} \frac{1}{\sigma\sqrt{2\pi}} exp\left(-\frac{a_k'^2}{2\sigma^2}\right) \right)
\tag{4}
$$

The main characteristics of the four presented models are summarized in Table 2.

### 4.4 Model learning

The above topic models (LDA, MCSM, gLDA and JMCSM) can be optimized using stochastic EM (SEM). In SEM, we alternately draw random signature assignment based on the current parameter estimation and then a new set of parameters is estimated given those assignments. As it is infeasible to directly calculate the signature assignment probability conditioned on the mutation categories, we use Gibbs sampling to randomly draw the assignments [27]. For

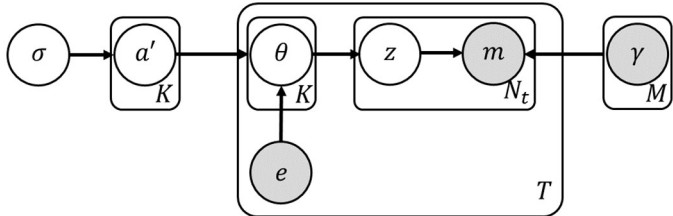

**Fig 6. A plate notation of gLDA.** The conditional distributions of each node are given by: $a'_k \sim Norm(0, \sigma^2)$; $\theta \sim Dir$ $(e \odot a)$; $z \sim Multi(\theta^I)$; and $m \sim Multi(\gamma_z)$.

instance, we execute Gibbs sampling for JMCSM by iteratively drawing assignments from the conditional probability (for the $n$th mutation of sample $t$):

$$p(z_n = k | w_n = m) \propto \frac{\{I_{k,t}\}_{\backslash n} + e_{kt}a_k}{I_t - 1 + \sum_k e_{kt}a_k} \gamma_{k,m} \tag{5}$$

where $\{I_{k,t}\}_{\backslash n}$ is the number of mutations drawn with assignment $k$, without counting the current mutation assignment. Given the sampled signature assignments, the model parameters are estimated using the L-BFGS-B optimzer. The process is stopped after a large number of iterations (3,000), tested for suffiency on generated data. Notably, when learning JMCSM we first extract $e_t$ by ignoring the mutation feature and applying the MMM model, optimized via EM.

## 4.5 Model evaluation

We evaluate the models by a held-out log-likelihood comparison with a feature oblivious analog: MCSM vs. LDA and JMCSM vs. gLDA. Since the likelihood of LDA and its derivatives is not directly computable, we follow previous works [28], and use the method of empirical likelihood (EL). In EL we draw a large number of tumor exposures using the estimated parameters, and then calculate the mean log-likelihood of the test set, marginalizing over all possible signature assignments per mutation. Specifically, we use $S = 10,000$ randomizations leading to $S$ pairs of exposure vectors for both feature values $(a_1^{(s)}, \ldots, a_K^{(s)})$ and $(b_1^{(s)}, \ldots, b_K^{(s)})$. The empirical log-likelihood of MCSM is given by (omitting a constant term that depends on $\sigma$):

$$\ell_{MCSM} = \frac{1}{S} \sum_{s=1}^{S} \sum_{t=1}^{T} \sum_{m=1}^{M} \left[ I_{t,m} \ln \left( \sum_{k=1}^{K} a_k^{(s)} \gamma_{km} \right) + J_{t,m} \ln \left( \sum_{k=1}^{K} b_k^{(s)} \gamma_{km} \right) \right] \tag{6}$$

For robustness, we run SEM for 50 iterations and report below the mean held-out EL calculated over the 25 last SEM iterations. Those numbers were chosen after examining generated data, observing that the held-out EL reaches a local maximum before the 25th iteration and then stays around it. Those results are summarized in Table C In S1 Text and Table D In S1 Text.

## 4.6 Data description

To test our novel models we work with Breast Cancer (BRCA), Malignant Lymphoma (MALY) and Chronic Lymphocytic Leukemia (CLLE) from International Cancer Genome Consortium as in [13] (see Table 3).

**Table 2. Main characteristics of the proposed probabilistic models.**

| Characteristic | LDA | MCSM | gLDA | JMCSM |
|---|---|---|---|---|
| Models mutation-level covariates | X | V | X | V |
| Utilizes inherent exposures | X | X | V | V |

**Table 3. Databases analyzed in this work.**

| Cancer Type | No. Samples | No. Mutations | Cosmic Signatures |
|---|---|---|---|
| BRCA | 560 | 3,472,652 | 1, 2, 3, 5, 6, 8, 13, 17, 18, 20, 26, 30 |
| MALY | 100 | 1,220,526 | 1, 2, 5, 9, 13, 17 |
| CLLE | 100 | 270,870 | 1, 2, 5, 9, 13 |

**Table 4. Number of mutations on each strand in the data sets we used.**

| Cancer Type | Lagging | Leading |
|---|---|---|
| BRCA | 313,219 | 209,488 |
| MALY | 96,177 | 68,365 |
| CLLE | 13,315 | 12,534 |

We executed 2-fold cross validation by dividing each sample into two equally-sized subsets. We learned the model parameters using one subset, and calculated the EL using the other one. To extract the inherent exposures, we used MMM on each sample of the train set separately. When using JMCSM and gLDA, we used the inherent exposures as observable variable to learn the model parameters. We also used the pre estimated inherent exposures to calculate the EL. We repeated this scheme twice to evaluate both MCSM and JMCSM compared with their strand-oblivious analogs. Data set sizes, restricted to mutations with replication strand information, are given in Table 4.

## Supporting information

**S1 Text. Fig A**. Radar plots showing reconstruction of MCSM model parameters using generated data. The angular axis is the signature number and the radial axis is its respective parameter value. We show that our method is able to recover the parameters in five different tests, and provide also the parameters estimated by LDA. **Fig B**. Radar plots showing reconstruction of JMCSM model parameters using generated data. Since we learn the strand bias per signature, we won't necessarily reconstruct the exact parameters. However, the ratio between the normalized parameters that indicates the bias of the signatures and was used in Fig 5, should be preserved instead. The radial axis indicates the ratio between the normalized parameters and the angular axis indicates the signature. We show that our method is able to reconstruct the bias trends in most cases, across five different tests. **Table A**. A comparison between the held-out log-likelihoods achieved by JMCSM and gLDA on generated data based on BRCA across five different tests. In all cases, JMCSM yields better results. **Table B**. A comparison between strand-sensitive and strand-oblivious models using genomic strand as a mutation-level feature. It is evident that in all instances that JMCSM and MCSM yield lower likelihoods than their strand indifferent variants. **Table C**. A summary of the number of iterations the algorithms ran on generated data before convergence, across five different tests. In all cases, the held-out log-likelihood achieved convergence before the 25th iteration. **Table D**. A summary of the number of iterations the algorithms ran on real data before convergence in all but one instance (when gLDA was executed on BRCA with the replication strand as a feature, it took 29 iterations). Although we report the mean of the held-out likelihood of the last 25 iterations, it doesn't have any effect on the results or the conclusions of the paper since the variations in the held-out loglikelihood are negligible relatively to the difference between the held-out loglikelihood achieved by JMCSM and gLDA.
(DOCX)

## Author Contributions

**Conceptualization:** Mark D. M. Leiserson, Roded Sharan.

**Data curation:** Mark D. M. Leiserson, Roded Sharan.

**Formal analysis:** Itay Kahane.

**Funding acquisition:** Mark D. M. Leiserson, Roded Sharan.

**Investigation:** Itay Kahane.

**Methodology:** Itay Kahane, Mark D. M. Leiserson, Roded Sharan.

**Project administration:** Roded Sharan.

**Resources:** Roded Sharan.

**Software:** Itay Kahane.

**Supervision:** Roded Sharan.

**Validation:** Roded Sharan.

**Visualization:** Itay Kahane, Roded Sharan.

**Writing – original draft:** Itay Kahane, Roded Sharan.

**Writing – review & editing:** Itay Kahane, Mark D. M. Leiserson, Roded Sharan.

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
