## [Decision Letter · Decision Letter 0]

7 Sep 2022

Dear Mr. Kahane,

Thank you very much for submitting your manuscript "A mutation-level covariate model for mutational signatures" for consideration at PLOS Computational Biology.

As with all papers reviewed by the journal, your manuscript was reviewed by members of the editorial board and by several independent reviewers. In light of the reviews (below this email), we would like to invite the resubmission of a significantly-revised version that takes into account the reviewers' comments.

The manuscript has now been reviewed by three RECOMB reviewers. While the reviewers have recognised the novelty of the work, they have expressed some concerns, especially related to a lack of clarity of the text and to the lacking experimental validation of the proposed methods. Therefore, we would like to invite the authors to submit a revised version of the manuscript which addresses the major concerns expressed by the reviewers, as well as a point-by-point response to address all their comments.

We cannot make any decision about publication until we have seen the revised manuscript and your response to the reviewers' comments. Your revised manuscript is also likely to be sent to reviewers for further evaluation.

Sincerely,

Simone Zaccaria

Guest Editor

PLOS Computational Biology

Jason Papin

Editor-in-Chief

PLOS Computational Biology

The manuscript has now been reviewed by three RECOMB reviewers. While the reviewers have recognised the novelty of the work, they have expressed some concerns, especially related to a lack of clarity of the text and to the lacking experimental validation of the proposed methods. Therefore, we would like to invite the authors to submit a revised version of the manuscript which addresses the major concerns expressed by the reviewers, as well as a point-by-point response to address all their comments.

Reviewer's Responses to Questions

**Comments to the Authors:**

**Reviewer #1**: 

Kahane et al. present a Latent Dirichlet Allocation model in which they incorporate covariates to observed categorised mutations in order to extract mutational signatures. Statistical methods for signature analysis are still lagging, so this work can be useful for linking signatures to their aetiology, using signatures in order to specify patients, etc.

The example they use of covariates is that of strand-specificity. Strand biases have already been assessed in COSMIC signatures, and are even already available in the COSMIC website (e.g. https://cancer.sanger.ac.uk/signatures/sbs/sbs2/). The authors should emphasize more strongly what is novel in their work.

The link to the covariates is done on parameter α′, i.e. the log-transformed Dirichlet parameter, and as such it determines both mean and variance of signature exposures, which should be made clearer.

The interpretation of results is on the normalised Dirichlet parameter α/sum α_i, and is compositional data. Therefore, signature-specific results are difficult and this needs to be made clearer in Figures 5 and 6.

The paper the authors claim to present a way of extracting signatures and exposures (e.g. Conclusions: We have developed new topic models to account for mutation-level covariates when learning mutational signatures and their exposures) However, they assume that mutational signatures are known and only estimate the exposures - which needs to be made clearer throughout the paper.

α′ is not defined in text.

I did not understand the held-out log-likelihood comparison method of assessing performance of the competing methods, please expand the explanation.

SBS2, SBS13 and SBS26 are put forward as signatures with strand bias. SBS13 does not appear to have strand bias according to https://cancer.sanger.ac.uk/signatures/sbs/sbs13. Please discuss.

**Reviewer #2:** 

This manuscript proposes two novel topic models for mutational signature learning. Unlike previous methods, these models explicitly account for the effect of covariates on the exposures to different signatures. The authors apply their methods to real data and conclude that they obtain a better fit to held-out data and that they identify known strand-specific biases, and find new ones.

We have several major concerns, one being that there are no simulation studies that prove that the proposed models and learning algorithms do what they set out to do. Additionally, the text is at times difficult to follow. We detail our concerns below.

The model description is unclear: what is a and b, and what is z drawn from? The symbols are introduced in the plate models before they are explicitly defined in the text, which makes this a very unpleasant read. In Figure 1, what is z drawn from? Is it a multinomial distribution with Nt and \\theta as parameters? And how does it generate m? And why is a_k generated from a Gaussian distribution?

In figures 1 and 2: The node for \\gamma is greyed out as an observed quantity, which seems odd as it is defined in the text as the probability to draw a mutation m from signature k.

In figures 3 and 4: The node for the inherent exposures is white, indicating it is an unobserved quantity. Later in the text, we see that it is inferred separately using a non-joint model, so in the scope of the joint model, it is given and so should be greyed out as an observed quantity.

Page 5: “we also define a guided LDA (gLDA) model variant which is fed with external information on the exposures as in JMCSM”. In the plate model this is an unshaded node, which means it is unobserved. But in the text, it is referred to as external information, which is confusing. Which is it?

In model learning, the authors should clarify what is the objective function. Are they computing the maximum marginal likelihood? If so, which parameters do they marginalize over and which do they optimize? And how are they including the prior information in the estimation? Similarly, it should be explained why model comparison based on the held-out log-likelihood is valid for the particular marginalization applied.

Additionally, the authors must describe their learning algorithms in more detail so that readers are able to understand the procedure and replicate it.

Page 6: “we run SEM for 50 iterations”. Since this is an EM algorithm, can’t the authors assess the convergence of the objective function directly?

Before applying their models to real data, the authors must benchmark them on simulated data, in order to see whether their learning method is working and whether they actually recover ground truth strand biases. Without such assessment, it is difficult to trust the results on real data.

The authors mention that the Watson/Crick strand information did not lead to good results, but show no data supporting this. They should support this statement with data.

**Reviewer #3: **

This paper introduces several probabilistic models and corresponding inference algorithms for inferring mutational signature exposures. These models are variations of the Latent Dirichlet Allocation model, capturing covariates among mutational signatures. This paper has been reviewed and accepted at RECOMB-CCB previously. Although I did not review this paper, a point-by-point response is missing, and the paper is still a bit rough (models are hard to read, typos, excessive number of acronyms, etc.). Moreover, the experimental evaluation is not great, lacking a compelling biological use case.

1. Improve description of the methods

I found the methods hard to read due to undefined variables, distributions and missing intuition. I will list some examples below.

- "The per-sample mutation category count of the two possible feature values are denoted I_{t,m} and J_{t,m}, respectively."

(1) What is m? -- from context one can guess it is a mutation category, but it would be good to be explicit about it.

(2) What is a single binary feature? What are the two possible feature values? Why only 2? Motivation is needed here.

- Why are a_k' ~ Norm(0, sigma^2)? This allows for negative values, whereas Dirichlet concentration parameters must be positive. I have seen gamma distributions being used as a prior for a_k.

- Figure 1 and Section 2.1. gamma is undefined -- do these describe signatures? What are the dimensions?

- Section 2.1: What is a signature count -- variable N_{t,k}?

- Figure 1: What are variables z and m?

- Figure 1: it would be good to include the conditional distribution for each node (with in-degree > 0). E.g. a_k' ~ N(0, sigma^2).

My above comments are for more the most basic model. Due to the lack of clarity for even this most basic model, it is hard to follow the other more complex models.

As a general comment, it would help to include a table, comparing the different models. Also, some additional motivation for each model would be welcome.

2. Experimental evaluation

My main concern with the current experimental evaluation is that the case for more complex models that support covariates has not been made. The experimental evaluation does not compare against a baseline method that does not include any covariates (this can be MMM, or just straight up NMF with the signature matrix fixed). Do more complex models allow for better inference with fewer mutations? The paper needs to make this case.

- I find it interesting that the "[paper] could detect no advantage of the models over their strand oblivious counterparts". Is this because the inference is more challenging for the more complex models? To assess this, simulations with known ground truth would be useful. Do the given algorithms for the more complex models have convergence issues?

- Equation (6) lists the empirical log-likelihood of the MCSM model. What about the other models?

- Do models with more parameters achieve higher empirical likelihoods by default?

Minor

* Figure 5, which model is this? MCSM or JMCSM?

* Figure 6 is not clear. Include some equations in the main text.

* Citation needed: "Signatures 2, 13, 26 are known to have a strong replication strand bias"

* dirichlet => Dirichlet

* signature => Signature

**Have the authors made all data and (if applicable) computational code underlying the findings in their manuscript fully available?**

Reviewer #1: Yes

Reviewer #2: Yes

Reviewer #3: None

PLOS authors have the option to publish the peer review history of their article (what does this mean?). If published, this will include your full peer review and any attached files.

Reviewer #1: No

Reviewer #2: No

Reviewer #3: No
---

## [Decision Letter · Decision Letter 1]

6 Dec 2022

Dear Mr. Kahane,

Thank you very much for submitting your manuscript "A mutation-level covariate model for mutational signatures" for consideration at PLOS Computational Biology. As with all papers reviewed by the journal, your manuscript was reviewed by members of the editorial board and by several independent reviewers. The reviewers appreciated the attention to an important topic. Based on the reviews, we are likely to accept this manuscript for publication, providing that you modify the manuscript according to the review recommendations. In particular, we ask the authors to please focus on addressing the remaining concerns, improvements, and clarifications requested by the reviewers.

Sincerely,

Simone Zaccaria

Guest Editor

PLOS Computational Biology

Jason Papin

Editor-in-Chief

PLOS Computational Biology

Reviewer's Responses to Questions

**Comments to the Authors:**

Reviewer #1: I had a very hard time linking the authors' response in the point-2-point reply to the changes made in the paper. I did not see any changes highlighted and often failed to identify any improvements. Here are some specific examples:

Authors: "The paper presents novel models that take mutation-level covariates into account, we have now emphasized the novel aspects in the Introduction."

However, advantage over eg [17] is not specified

Last review: "The link to the covariates is done on parameter α′, i.e. the log-transformed Dirichlet parameter, and as such it determines both mean and variance of signature exposures, which should be made clearer."

Authors: "We have now clarified the way the covariate affect the model (p. 4)."

However, p4 does not make any reference to α′

Last review: "The interpretation of results is on the normalised Dirichlet parameter α/sum α_i, and is compositional data. Therefore, signature-specific results are difficult and this needs to be made clearer in Figures 5 and 6."

Authors: "We have now explained more clearly the caveats of the parameter ratio approach in Figure 5 and how it is addressed in the results of Figure 6 to pinpoint the relevant biased signatures (p. 10)."

However, p10 does not make any reference to interpretation of parameters

Last review: "The paper the authors claim to present a way of extracting signatures and exposures (e.g. Conclusions: We have developed new topic models to account for mutation-level covariates when learning mutational signatures and their exposures) However, they assume that mutational signatures are known and only estimate the exposures - which needs to be made clearer throughout the paper."

Authors: "We have stated this point (of known signatures) more clearly (p. 3,11)."

I could find this change. However, it needs to be clarified if the authors mean “known signatures” by “inherent exposures”?

Last review: "α′ is not defined in text."

Authors: "We added an explicit definition of the a_i and a' parameters (p. 3)"

However, I could not find it on p. 3

These discrepancies between the revised manuscript and the P2P are very surprising and I wonder if I have been sent the correct documents.

Reviewer #2: The authors have responded to most of our comments, but most concerns have not been addressed satisfactorily. Even though they improved the exposition problems we and other reviewers mentioned, the algorithm description is still not clear enough. Additionally, we still have doubts about the convergence assessment and the simulation results. We outline these concerns below.

The authors have provided us (but not the main text) with tables showing the progress in held-out log-likelihood of the different models during the Stochastic EM algorithm in order to show that 50 iterations are enough. However, not only do some runs not converge, as they point out, but they also miss the main point of our comment: the method should not be run with a fixed number of iterations for all data sets, as it is natural that the time to convergence varies for different data sets. Instead, they should run their method for as many iterations as it needs to achieve convergence (within some computational budget), based on the progression of the log-likelihood.

The authors have also provided us (but not the main text) with some simulation results. The authors assess the ability of their method to recover the simulated parameters from the simulated data and compare them with a random estimator. Firstly, they should clarify what the “random estimator” is. Secondly, they should show their results in a visual way by using boxplots aggregated across different levels of difficulty in the simulations. Thirdly, and most importantly, the goal of the simulations is to show the performance of each method to recover the simulated true quantities of interest and compare them. Specifically, if the point is to show that the joint model is able to capture strand specificity, the authors must show that it indeed recovers them better than the alternative baseline.

Additionally, just comparing the held-out likelihoods between the different models is not enough to motivate the more complex ones. The question that the authors must answer unequivocally in their paper using simulation and real data results is whether the estimated exposures more closely capture the ground truth using the proposed methods, instead of the simpler alternatives.

Finally, the code provided in the Github repository is insufficient to reproduce the results in the paper, as it is simply a collection of compressed scripts and does not include instructions on how to run the models on new data.

**Have the authors made all data and (if applicable) computational code underlying the findings in their manuscript fully available?**

Reviewer #1: Yes

Reviewer #2: Yes

Reviewer #3: None

Figure Files:

Data Requirements:

Reproducibility:

References:

---

## [Decision Letter · Decision Letter 2]

23 Mar 2023

Dear Mr. Kahane,

Thank you very much for submitting your manuscript "A mutation-level covariate model for mutational signatures" for consideration at PLOS Computational Biology. As with all papers reviewed by the journal, your manuscript was reviewed by members of the editorial board and by several independent reviewers. The reviewers appreciated the attention to an important topic. Based on the reviews, we are likely to accept this manuscript for publication, providing that you modify the manuscript according to the review recommendations.

While most of the reviewers are now satisfied with the provided answer, one last reviewer believes that their comments have not been properly addressed and adding some proof-of-concept simultations would be essential to finalise the statements in the manuscript. Therefore, we would like to ask the authors to please consider this last comment from the reviewer and add some basic proof-of-concept simulations including strand-specific effects to address this remaining concern. Also, we would like to ask the authors to please make sure that the code to reproduce the analysis will now be made publicly available and a corresponding link should be provided in the text for review.

Sincerely,

Simone Zaccaria

Guest Editor

PLOS Computational Biology

Jason Papin

Editor-in-Chief

PLOS Computational Biology

Reviewer's Responses to Questions

**Comments to the Authors:**

Reviewer #2: Unfortunately, the authors still seem to miss our point about assessing their methods’ ability to obtain the true parameters from simulated data. In order to illustrate the advantages of the strand-specific models, the authors must simulate data sets with different levels of strand-specific effects, and assess the ability of each method to recover the simulated exposures. They must then report some metric that summarizes this performance across many instances of each simulation case, instead of plotting the learned parameter values on top of the true values for each instance of the simulation. For example, they may report the error between the true versus estimated exposures and show the distributions of these errors for each method, and each simulation setting across all replicates.

Additionally, while the authors did improve their simulation study, its description is still very sparse and not reproducible. The authors must be clear and detailed about their procedures for simulating and evaluating the methods on those simulations. For example, “Then, the data was split to a train set and a test set.” is not enough. Please provide the actual splitting procedure.

They also explained in their rebuttal what the “random estimator” is, but did not add that information to the manuscript. And the small simulation results they show in the review are still not added to the main text.

Finally, while we do agree that the plot indicates that convergence is reached well before 50 iterations in that particular case, there is no reason to believe that this will generalise to other datasets. We advise to define a more general convergence criterion as is commonly done.

**Have the authors made all data and (if applicable) computational code underlying the findings in their manuscript fully available?**

Reviewer #2: **No: **Code for simulation is not available; thus not all results are reproducible.

PLOS authors have the option to publish the peer review history of their article (what does this mean?). If published, this will include your full peer review and any attached files.

Reviewer #2: No

Reviewer #3: No

Figure Files:

Data Requirements:

Reproducibility:

References:

---

## [Editor Report · Decision Letter 3]

17 May 2023

Dear Mr. Kahane,

We are pleased to inform you that your manuscript 'A mutation-level covariate model for mutational signatures' has been provisionally accepted for publication in PLOS Computational Biology.

Best regards,

Simone Zaccaria

Guest Editor

PLOS Computational Biology

Jason Papin

Editor-in-Chief

PLOS Computational Biology

---

## [Editor Report · Acceptance letter]

1 Jun 2023

PCOMPBIOL-D-22-01010R3 

A mutation-level covariate model for mutational signatures

Dear Dr Kahane,

I am pleased to inform you that your manuscript has been formally accepted for publication in PLOS Computational Biology. Your manuscript is now with our production department and you will be notified of the publication date in due course.

With kind regards,

Zsofia Freund
